# A Novel Multi-Scaled Deep Convolutional Structure for Punctilious Human Gait Authentication

**DOI:** 10.3390/biomimetics9060364

**Published:** 2024-06-16

**Authors:** Reem N. Yousef, Mohamed Maher Ata, Amr E. Eldin Rashed, Mahmoud Badawy, Mostafa A. Elhosseini, Waleed M. Bahgat

**Affiliations:** 1Delta Higher Institute for Engineering and Technology, Mansoura 35681, Egypt; reem.nehad@hotmail.com; 2School of Computational Sciences and Artificial Intelligence (CSAI), Zewail City of Science and Technology, October Gardens, 6th of October City, Giza 12578, Egypt; momaher@zewailcity.edu.eg; 3Department of Communications and Electronics Engineering, MISR Higher Institute for Engineering and Technology, Mansoura 35516, Egypt; 4Department of Computer Engineering, College of Computers and Information Technology, Taif University, Taif P.O. Box 11099, Saudi Arabia; a.rashed@tu.edu.sa; 5Department of Computer Science and Informatics, Taibah University, Medina 42353, Saudi Arabia; wshehata@taibahu.edu.sa; 6College of Computer Science and Engineering, Taibah University, Yanbu 46421, Saudi Arabia; 7Department of Computers and Control Systems Engineering, Faculty of Engineering, Mansoura University, Mansoura 35516, Egypt; melhosseini@mans.edu.eg; 8Information Technology Department, Faculty of Computers and Information, Mansoura University, El Mansoura 35516, Egypt

**Keywords:** CASIA gait dataset, convolutional neural network (CNN), MLP, OU-ISIR, silhouette images

## Abstract

The need for non-interactive human recognition systems to ensure safe isolation between users and biometric equipment has been exposed by the COVID-19 pandemic. This study introduces a novel Multi-Scaled Deep Convolutional Structure for Punctilious Human Gait Authentication (MSDCS-PHGA). The proposed MSDCS-PHGA involves segmenting, preprocessing, and resizing silhouette images into three scales. Gait features are extracted from these multi-scale images using custom convolutional layers and fused to form an integrated feature set. This multi-scaled deep convolutional approach demonstrates its efficacy in gait recognition by significantly enhancing accuracy. The proposed convolutional neural network (CNN) architecture is assessed using three benchmark datasets: CASIA, OU-ISIR, and OU-MVLP. Moreover, the proposed model is evaluated against other pre-trained models using key performance metrics such as precision, accuracy, sensitivity, specificity, and training time. The results indicate that the proposed deep CNN model outperforms existing models focused on human gait. Notably, it achieves an accuracy of approximately 99.9% for both the CASIA and OU-ISIR datasets and 99.8% for the OU-MVLP dataset while maintaining a minimal training time of around 3 min.

## 1. Introduction

Over recent decades, research in biometric identification has progressively advanced, focusing on non-contact methods [1]. Biometrics, which identify individuals based on biological and behavioral traits [2], encompass various forms, including fingerprint [3], DNA [4], facial recognition [5], iris scanning [6], and gait recognition [7]. Gait recognition has been a prominent method in the growing demand for intelligent and secure monitoring, demonstrated most notably during the COVID-19 epidemic [8]. Gait recognition is distinguished among biometric systems due to its inherent non-contact nature. This characteristic allows the identification process to be conducted from a distance without requiring the subject’s direct interaction with the biometric capturing device. The non-invasive and hygienic nature of gait recognition is particularly advantageous, as highlighted during scenarios such as the COVID-19 pandemic. This capability to operate effectively without physical contact places gait recognition in a unique position compared to other biometric methods that necessitate proximity or direct interaction [9]. It is also resilient to attempts at disguise or concealment. It functions well with low-resolution images, making it versatile for various security contexts, such as shopping centers, healthcare facilities, financial institutions, airports, and military installations.

The gait identification models are essential in biometric identification via analyzing walking patterns. There are two main types: Model-Based, focusing on body part analysis such as joints, and Model-Free, which uses binary isolated images or gait energy images (GEIs). These models contribute uniquely to recognizing and understanding individual gait patterns, each with distinct methodologies and applications.

The Model-Based gait recognition approach involves analyzing body parts, such as a participant’s joints, to develop a gait recognition algorithm. High-quality input frames are crucial for their effectiveness, typically necessitating a multi-camera setup for capturing detailed gait sequences [10]. Model-Based algorithms are versatile, handling variations like changes in clothing or the presence of carried objects. However, challenges in task recognition and the high cost of participant modeling limit its practical popularity [11].

Model-Free gait recognition is an appearance-based method that utilizes binary isolated images for gait recognition. It is notably applied in two ways: (i) using human gait silhouette images for datasets like CASIA and OU-ISIR and (ii) employing gait energy images (GEIs) for the OU-MVLP dataset [11]. The GEI approach is preferred for its low computational cost and high recognition rate, achieved by aligning and averaging silhouettes and comparing successive GEIs. Alternatively, the silhouette-based method inputs silhouette images directly into deep convolutional neural networks (CNNs) for feature extraction. Compared to GEI, this method further improves recognition by leveraging the advanced capabilities of CNNs [12].

When it comes to the difficulties of gait identification, deep convolutional neural network (CNN) methods have proven to be beneficial. Additionally, deep learning-based gait authentication has applications in human psychology, such as analyzing emotional patterns through point-light displays [13].

This study introduces a new approach to human gait authentication: the Multi-Scaled Deep Convolutional Structure for Punctilious Gait Analysis (MSDCS-PHGA). This study’s primary contributions include the following:Introducing a novel technique for extracting and resizing gait images into three scales;Developing a deep CNN algorithm specifically tailored for efficient gait feature extraction;An innovative approach for the fusion of features extracted at multiple scales enhances the robustness of gait recognition;Implementing a fine-tuned multilayer perceptron (MLP) for feature recognition;Developing an optimized convolutional neural network (CNN) model;Conducting a comprehensive comparative analysis with leading traditional CNN algorithms demonstrates that the suggested model is more accurate and efficient than state-of-the-art gait recognition technology.

The paper is organized into six sections: Section 2 presents related works, Section 3 describes the CNN models, Section 4 details the methodology, Section 5 discusses the results, and Section 6 concludes the paper and introduces future research directions.

## 2. Related Works

This section presents an overview of current developments in gait recognition, exploring approaches ranging from traditional models to advanced CNN techniques. It highlights the progression of gait recognition technology, detailing methods that enhance accuracy and expand applications.

Altilio et al. [14] presented a reliable monitoring scheme to automatically record and classify patient movements based on machine learning classification methods. The largest accuracy value scored 91% in affordable and residence-based rehabilitation methods from Probabilistic Neural Networks (PNNs). Using the suggested deep convolutional neural network (DCNN), Saleh et al. [15] presented a person recognition algorithm. Once Image Augmentation (IA) had been applied to the identical data, its performance indicators were compared. The trial outcomes scored 96% with IA compared to 82% without IA. Liao et al. [11] introduced a gait authentication model named PoseGait, which utilizes spatiotemporal information extracted from poses to improve gait identification rates. This study demonstrated that the suggested system has a more reliable performance.

Moreover, Liao et al. [16] have developed a new Dense-vision GAN (DV-GAN) model that addresses the problem of large-angle intervals in vision synthesis by utilizing view space covering. The modeling procedure of DV-GAN enhanced the discriminative performance of synthesized gait images. Subsequently, the characteristics of the model will enhance the gait recognition model.

Elharrouss et al. [17] introduced a gait identification model that utilizes a deep CNN to estimate and recognize the angle of the gait. The proposed model scored more than 98% recognition accuracy for three gait datasets. Zou et al. [18] developed a gait authentication technique based on a hybrid deep neural network combining a recurrent neural network (RNN) and a CNN. This model has trained two datasets collected by smartphones from 118 subjects in the wild. The experimental results achieved more than 93% in-person identification accuracy. Wu et al. [19] presented the gait recognition system by extracting discriminative features from video sequences’ temporal and spatial domains. The proposed method has identified the correlation between video frames and the temporal variation in a specific region. The findings have demonstrated a significant enhancement in gait identification. Guo et al. [20] employed the Gabor Filter technique to extract gait features from gait energy images. Subsequently, they utilized Linear Discriminant Analysis (LDA) to address the issue of feature size. The gait features were ultimately categorized using the Particle Swarm Optimization Algorithm (PSA). The results indicate that the suggested recognition algorithm achieves flawless performance metrics in cross-view gait recognition.

Aybuke et al. [21] tested ten standard machine learning algorithms on the data gait called (HugaDB) of 18 individuals. The empirical findings demonstrated superior performance in the IB1, Bayesian Net, and Random Forest (RF) algorithms. Zhang et al. [22] have presented a novel recognition algorithm called Joint Unique-gait and Cross-gait Network (JUCNet) to improve the gait recognition performance. Extensive tests revealed that the proposed method outperformed all other studies. M. Sivarathinabala and S. Abirami [23] extracted static and dynamic features from CMU motion capture and the TUMIIT KGP gait database and then applied feature fusion between both. Finally, the experimented result scored 97% efficiency when the fused features were recognized based on the support vector machine model.

Saleem et al. [24] employed deep learning techniques to identify the CASIA-B gait database. A comprehensive framework was implemented to incorporate feature extraction, selection, and classification of the fused features utilizing various methodologies. The gait features have been derived from the enhanced data using pre-trained models, specifically Inception-ResNet-V2 and NASNet Mobile. By employing the modified mean absolute deviation extended serial fusion (MDeSF) method, the amalgamation of the most optimal characteristics yielded an experimental outcome with an accuracy of 89%. Shiqi et al. [25] proposed a model called GaitGANv2 using generative adversarial networks (GANs). The presented algorithm varies from classic GAN in that GaitGANv2 has two discriminators rather than one. GaitGANv2 has attained state-of-the-art performance, according to experimental results.

E. Anbalagan et al. [26] proposed a gait recognition model by extracting features from a gray-level co-occurrence matrix (GLCM) from the preprocessed images. Then, the Long Short-Term Memory (LSTM), multilayer perceptron (MLP), and deep neural networks (DNNs) were classified as images from the extracted features. Finally, the experimental results scored a 96% accuracy with a 0.04% error value.

In recent advancements in gait recognition, self-supervised learning techniques have emerged as powerful tools for extracting meaningful representations from unlabeled data. Specifically, self-supervised contrastive learning has been effectively utilized to learn invariant gait representations, which are crucial for predicting the freezing of gait events in patients with Parkinson’s disease [27]. This approach leverages time series acceleration data to generate robust, class-specific gait representations without the reliance on labeled datasets. By employing contrastive learning, the model captures essential gait features and patterns, enhancing the prediction accuracy and generalizability across various gait analysis tasks.

Contrastive learning’s effectiveness extends beyond traditional gait recognition, contributing significantly to biosignal analysis and computer vision. The self-supervised nature of this learning paradigm allows for more efficient model training and adaptation in scenarios where labeled data are scarce or unavailable, thereby improving the robustness of gait recognition systems.

This contrasts with supervised learning approaches where models are trained with labeled data, such as using artificial neural networks (ANNs) to enhance the accuracy of estimations made by robot-mounted 3D cameras in human gait analysis [28]. While supervised methods rely on high-quality, labeled datasets to achieve high accuracy, self-supervised methods like contrastive learning offer a viable alternative for improving system performance in less constrained environments.

Wang et al. [29] presented a gait identification model by a convolutional LSTM approach called Conv-LSTM. They took a few steps to execute the gait recognition procedure. Firstly, they extracted silhouette images from video frames to create gait energy images (GEIs) from them and then enlarged their volume to relieve the gait cycle’s limitation. Later on, they experimented with examining one subject’s cross-covariance. Finally, they accomplish gait recognition by creating a Conv-LSTM model. The experimental results scored 93% and 95% accuracy on CASIA-B and OU-ISIR, respectively. Zhao et al. [30] addressed the issue of multiview gait identification, specifically the lack of consistency across different viewing angles, by introducing a spiderweb graph neural network (SpiderNet). This network connects a single view and other view angles concurrently. Their suggested methodology was implemented on three datasets called CASIA-B, OU-MVLP, and SDUgait and scored an accuracy of 98%, 96%, and 98%, respectively.

Each of these studies contributes to the evolving landscape of gait recognition technology. Table 1 illustrates a comparative study of the various state-of-the-art techniques for gait recognition. In the field of gait recognition, despite advancements using deep learning techniques, several research gaps remain that need addressing to enhance the robustness and applicability of these systems in real-world conditions:Real-World Data Application: There is a lack of models rigorously tested on diverse, real-world datasets. Improvements are needed for systems to perform well in various environmental and situational contexts;Viewing Angle Variability: Many models struggle with large variations in viewing angles. Enhanced view synthesis methods could help improve recognition accuracy in practical settings;Dataset Quality: Enhancements in data segmentation and dataset quality are necessary to improve model training outcomes and recognition accuracy;System Complexity and Accuracy: Some innovative models exhibit low accuracy due to their complexity. Simplifying these models while maintaining or improving accuracy is crucial;Cross-View Recognition: Improved techniques for cross-view gait recognition are needed to ensure consistent performance across different viewing angles.

Addressing these gaps involves developing more adaptable models and constructing datasets that better mirror the complexity of everyday scenarios.

## 3. Convolutional Neural Network Models (CNNs)

Convolutional neural networks (CNNs) are a class of multilayer artificial neural networks or deep learning (DL) architecture that draws inspiration from the visual systems of real creatures [35]. Recently, CNNs have gained popularity in image processing as they improve performance without extracting features from segmented pictures [36]. These networks are utilized in various fields, such as visual tasks, natural language processing, image identification, and classification [37]. A typical convolutional neural network (CNN) comprises layers that perform linear and nonlinear operations, which are acquired jointly [38]. The essential components of a convolutional neural network (CNN) consist of convolution layers, which extract features; pooling layers, which perform down-sampling operations on the feature maps; and fully linked layers at the end to flatten these maps [39]. Other CNN designs are available, including LeNet, AlexNet, VGG, Inception, ResNet, and Xception [35]. Although each model has a distinct structure, they share foundational elements.

## 4. Methodology

This section introduces the proposed CNN models for gait recognition, beginning with the proposed CNN model, which intends to improve gait recognition. The model also seeks to expand the scope of gait applications, such as tracking the spread of COVID-19 [15], rather than other widely used biometrics, like fingerprints, faces, iris, and DNA. The section also covers the CASIA gait [40], OU-ISIR [41], and OU-MVLP [33,42] datasets used for pre-training these models. Moreover, it outlines the preprocessing techniques to enhance image quality and prepare the data for effective analysis.

The conceptual design of the proposed Multi-Scaled Deep Convolutional Structure for Punctilious Human Gait Authentication (MSDCS-PHGA) is foundational to understanding the innovative approach this study introduces to gait recognition. The MSDCS-PHGA architecture is engineered to efficiently process and analyze human gait data captured in silhouette form. The gait authentication approach, detailed in Figure 1, employs CNNs to process silhouette images at three scale resolutions: 50 × 50, 100 × 100, and 150 × 150 pixels. Each set of images undergoes processing by a dedicated CNN, with the resulting features fused for classification by a fully connected network comprising an input layer, a dense hidden layer, and an output layer with four neurons.

Figure 1 illustrates the detailed workflow of MSDCS-PHGA, showing the process from image capture to final gait authentication. Each component of the conceptual design is mapped to its operational counterpart in the system architecture, providing a clear visualization of the method’s functionality. It consists of three core components: multi-scale image processing, feature extraction through convolutional layers, and feature fusion for final gait classification as follows:Multi-Scale Image Processing: To accommodate variations in distance and perspective that naturally occur in gait data, silhouette images are segmented, preprocessed, and resized into three distinct scales: 50 × 50, 100 × 100, and 150 × 150 pixels. This scaling ensures that the network learns to recognize features across different resolutions, enhancing its ability to generalize across diverse real-world conditions;Feature Extraction: Each scaled image is processed by a custom-designed convolutional neural network (CNN). These networks are tailored to extract spatial hierarchies of features from simple edges to more complex gait patterns. The architecture of each CNN layer is specifically optimized to maximize the extraction of discriminative features from the gait data, which is crucial for the subsequent classification accuracy;Feature Fusion: After feature extraction, a fusion mechanism is employed to integrate the features from all scales into a coherent feature set. This integrated set harnesses the complementary information available across different scales, significantly boosting the robustness and accuracy of the gait recognition process.

The MSDCS-PHGA is grounded in the theory that a deeper analysis of spatial features at multiple scales can substantially improve gait recognition accuracy. By harnessing multi-scale data, the model effectively captures a more comprehensive range of biomechanical and behavioral gait dynamics, often lost in single-scale approaches. The proposed design integrates seamlessly with existing biometric systems, enhancing reliability without requiring extensive modifications. It is compatible with standard image capture devices and can be easily implemented alongside existing security protocols, significantly improving non-intrusive biometric authentication.

### 4.1. The Proposed Procedures for Gait Recognition

In this study, all algorithms for gait recognition follow a comprehensive five-step procedure. Each stage of this procedure is crucial, contributing significantly to the overall effectiveness of the process. Step 1 involves preprocessing and segmenting all gait datasets, where silhouettes and gait energy images (GEIs) are extracted as the primary data forms, per the guidelines in reference [43]. This step ensures that the data are refined and suitable for detailed analysis. The second step is dedicated to extracting distinct gait features from these prepared multi-scale gait images, capturing the unique aspects of individual gaits.

In Step 3, the focus is on concatenating and fusing the extracted features from the multi-scale images, creating a comprehensive and robust feature set to address variations in gait patterns effectively. Step 4 involves training and testing deep learning modules, encompassing a range of traditional and traditional algorithms proposed in this research. This phase aligns them with the latest advances in gait recognition technology.

The final step, Step 5, is the computation and analysis of key performance metrics such as precision, F-score, recall, False Negative Rate, and training time, providing a quantitative assessment of the algorithms’ effectiveness. The procedural intricacies of these steps are systematically outlined in Algorithm 1. To complement the textual explanation, Figure 2 visually illustrates the main steps of the proposed model’s flowchart [44]. Figure 3 provides an illustrative example of a gait energy image (GEI) for an individual, showcasing the data analyzed.
**Algorithm 1: Gait Authentication Model**
**Input:** Isolated Gait Dataset (*I*).
**Output:** Identified gait Images.
//Read the Isolated Gait dataset (ROI)**1.***I ← read(ROI)*
// Normalize all isolated images (I)**2.***N ← normalize(I)*
//Split images (I) for training, testing, and validation by (Train_test_split)**3.***train_X, test_X, valid_X, train_y, test_y, valid_y ← split(N)*
//Training and testing Algorithms**4.***M ← [ABDGNet, LeNet, AlexNet, VGG, Inception, ResNet50, Xception]*
//Extract Features from three models.**5.****Foreach** *model m in M***6.**
*F ← extract(m, train_X, test_X, valid_X)* // Extract Features**7.***F_fused_ ← concatenate(F)***8.***m_Fused_ ← train(F_fused_, train_y)* //Apply Concatenation Feature Fusion**9.***Summary ← m_Fused_* //Train the Fused Model Layers**10.***m_Compiled_ ← compile(m_Fused_)* //Summary of the Fused Model**11.***epochs ← 30***12.***batch_size ← 32***13.***ver ← 2***14.***m_Fit_ ← fit(m_Compiled_, train_X, train_y, epochs, batch_size, ver)***15.***plot(m_Fit_)***16.***ConfMatrix ← calculate(m_Fit_, test_X, test_y)***17.***plot(ConfMatrix)***18.***Store ← (m_Fused_)* //Store the Model**19.****End For**

### 4.2. Dataset Description

CASIA gait datasets [40] were created by the Institute of Automation Chinese Academy of Sciences and included four gait datasets: A, B, C, and D. Moreover, the Institute of Scientific and Industrial Research (ISIR) has presented OU-ISIR, which contained three gait datasets: A, B, and OU-MVLP. The proposed approach is applied to CASIA, OU-ISIR, and OU-MVLP gait datasets, which included different categories for study. The total number of images in the three applied datasets is around 20 K, where each one is different owing to several view angles, speeds, and clothing variations. Table 2 shows the gait dataset description, and Figure 4 depicts a selection of the applied datasets.

#### 4.2.1. CASIA

Dataset A has taken from 20 persons. Each individual has 12 image patterns, and every four contains three directions to the image surface, which are 0°, 45°, and 90°. Each sequence has a unique length depending on the walker’s speed, but it must range between 37 and 27. This dataset contains 19,139 images in a size of 2.2 GB [20,31]. Dataset B is a large multi-view gait dataset created in January 2005. It was collected from 124 participants, and the gait data were collected from 11 different view angles, ranging from 0° to 180° divided by 18° with three different variations: walking manner, carrying, and clothing condition changes [45,46]. Dataset C was created in August 2005 with an infrared camera. It collected 153 participants in four walking conditions (slow walking, normal walking with or without the bag, and fast walking). These videos were all taken at night [31].

In this study, we prioritized the privacy of participants. To comply with ethical standards, we anonymized the images by cropping out the faces of individuals. This modification made post-experimentation does not impact the scientific validity of the results, as it solely involves the facial regions, leaving the key aspects of gait analysis intact. This step ensures participant confidentiality while maintaining the integrity of our research findings.

#### 4.2.2. OU-ISIR

The OU-ISIR dataset was created in March 2007 [41], which contains 4007 subjects (1872 females and 2135 males) with ages varying from 1 to 94 years. It was gathered from treadmill subjects surrounded by 25 cameras at 60 frames per second, 640 by 480 pixels [32]. The OU-ISIR dataset contains two subsets: 9 speed variations between 2 and 10 km/h in OU-ISIR (A) and garment variations up to 32 combinations in OU-ISIR (B) datasets [41].

#### 4.2.3. OU-MVLP

The Multi-view Large Population Dataset (OU-MVLP) was created in Feb. 2018 [42] and collected from 10,307 subjects (5114 males and 5193 females) with different ages from 2 to 87 years old. This dataset contains 14 view angles spanning from 0° to 90° and 180° to 270°, taken by seven cameras (Cam1–7) mounted at 15-degree azimuth angles along a quarter of a circle whose center coincides with the walking course’s center.

### 4.3. Preprocessing Dataset

Image preprocessing procedures are used to improve the image quality. Preprocessing aims to reduce noise, remove distortion, and eventually suppress or highlight other attributes essential for subsequent processing, such as segmentation and recognition. Firstly, the gait dataset images were scaled down to three different sizes: 150 × 150, 100 × 100, and 50 × 50, then converted to gray-scaled images to reduce the computational time. In addition, subtract each gait image from its background to obtain isolated images and apply histogram equalization to contrast the resulting image, which is expressed by Equation (1) [47]:(1)$$Y=\left\{f\left(x\left(i,j\right)\right)|\forall x\left(i,j\right)\in X\right\}$$

The segmented images are then converted to binary images by the Otsu threshold approach [48], which maximizes the between-class variance, as computed by Equations (2) and (3):(2)$$\sigma_{B}^{2}=\omega_{0}{\left(\mu_{0}-\mu_{T}\right)}^{2}+\omega_{1}{\left(\mu_{1}-\mu_{T}\right)}^{2}$$
(3)$$t^{*}={argmax}_{1\leq t\leq L}\sigma_{B}^{2}$$
where $\omega_{0}\;and\;\omega_{1}$ are referred to as the probabilities of the foreground and background parts. Also, ${\;\mu }_{0},\;\mu_{1}$, and $\mu_{T}$ are represented as the mean of the gray level of the foreground, background, and entire gray-level image, respectively.

Finally, the binary images are passed through the morphological operations of dilation and filling to remove the impurities from the isolated binary images using a disk Structure Element (SE) with a radius of 1. In addition, all isolated images have normalized for enhancing images by creating a new range from an existing one [49]. The dilation of an image A (set) by structuring element B is defined as Equation (4) [50]:(4)$$A⊕B=\left\{z|{\left(Ḃ\right)}_{z}\cap A\neq \emptyset \right\}$$

The filling operation has been predicated on filling the entire region with ‘black’ beginning at any point within the boundary, which is computed by Equation (5) [51]:(5)$$X_{K}={(X}_{K-1}⊕B)\cap A^{c}\;\;\;\;\;\;\;\;\;\;\;\;\;\;\;\;\;K=1,2,3,\ldots \ldots .$$
where ${\;X}_{0}$ = P, B is the Structure Element (SE), and *A^c^* is the complement of set A. Figure 5 refers to the proposed gait preprocessing procedures applied in the proposed study. Figure 6 depicts a sample of the preprocessed datasets.

The dataset can be normalized by a predefined boundary, which can be calculated by Equation (6) [49]:(6)$$Á=\left(\frac{A-\min value\;of\;A}{\max value\;of\;A-\min value\;of\;A}\right)*\left(D-C\right)+C\;$$

Á contains Min–Max normalized data one when predefined boundaries are [C, D] and A is the main data range. Algorithm 2 presents the main preprocessing procedures of the gait images.
**Algorithm 2: Preprocessing Steps**
**Input:** RGB gait dataset (*D*) and Background images (*D_Back_*).
**Output:** binary isolated gait dataset images (*I*).*1.***Procedure** Generate Isolated Gait Dataset (*D, D_Back_*):
// Resize all images to three sizes (150, 100, and 50)*2.*
*D_Resized_**←**resize(D, [150, 100, 50])*
// Apply Rgb2gray image conversion*3.**D_GS_**←**grayScale(D_Resized_)*
// Apply subtraction between grayscale gait and background*4.**I**←**D_GS_ – D_Back_*
// Apply histogram equalization*5.**I_Equalized_**←**histogramEqualization(I)*
// Apply the image Thresholding procedure*6.**I_Threshold_**←**OtsuThresholding(I_Equalized_)*
// Construct a Structuring Element with disk radius = 1*7.**SE**←**CreateStructuringElement (radius = 1)*
// Dilate image*8.**I_Dilated_**←**DilateImage(I_Threshold_, SE)*
// Fill the holes*9.**I_Filled_**←**fillHoles(I_Dilated_, threshold = t)*
// Remove all connected components less than t*10.**I**←**RemoveSmallComponents(I_Filled_, threshold = t)**11.***Return***I**12.***End Procedure**

### 4.4. The Structure of the Proposed CNN Model

After processing and isolating the images from the gait datasets, a novel deep convolutional neural network, Appearance-Based Deep Gait Network (ABDGNet), is constructed for gait image recognition. The architecture of ABDGNet is meticulously designed and consists of four convolutional layers. Following each convolutional layer is a max pooling layer with a 2 × 2 mask size.

The first two convolutional layers are equipped with 32 filters, each 3 × 3 in size, with a stride of 1, and employ the ReLU activation function for nonlinear processing. The subsequent two layers are more complex, each containing 64 filters of 3 × 3 in size, but with a stride of 2, and also utilize the ReLU activation function.

The ReLU classifier carries out the classification in ABDGNet, the output of which is computed using Equation (7) [52]:(7)$$ý=arg\;max\;\;max\left(0,o\right)$$
i∈1,……………,N

In the subsequent stages, the output is flattened to transition from convolutional to fully connected layers. The network includes three dense layers with 1024, 512, and 256 depths, respectively. After the multi-scale features are concatenated, they pass through another dense layer with a depth of 100, followed by a SoftMax layer for final classification, as defined by Equation (8):(8)$$o=Softmax\left({feat}_{full}*W_{o}+b_{o}\right)$$
where $W_{o}$ is the weight matrix of the output layer and $b_{o}$ is the bias of the output layer. The output of each layer can be calculated by Equations (9) and (10) [53].
(9)$$L_{output}=\left[\frac{N-F+2P}{S}\right]+1$$

Or
(10)$$L_{output}=\left[\frac{N-F}{S}\right]+1$$

N is the input size, *F* refers to the filter number, *P* is the padding size, and *S* is the stride value. Equation (9) computes the output layer with padding value, and Equation (10) computes the output in case of zero padding. Figure 7 presents the overall architecture of ABDGNet. Figure 8 illustrates the detailed layers of ABDGNet using the CASIA dataset as a sample.

### 4.5. Performance Evaluation

The performance metrics of the model are thoroughly evaluated using data from the confusion matrix. Essential parameters derived from the confusion matrix include the true positive value (TP), the true negative value (TN), the false positive value (FP), and the false negative value (FN). Figure 9 illustrates a sample confusion matrix for the proposed system using the CASIA-B dataset. The model’s performance metrics evaluation encompasses accuracy, sensitivity, specificity, precision, false discovery rate, F1-score, training time, and recall rate (R) [47]. The accuracy of ABDGNet is calculated using Equation (11) [54]:(11)$$Accuracy=\frac{TP+TN}{TP+TN+FP+FN}\times 100$$

The sensitivity or True Positive Rate (TPR) and the specificity or True Negative Rate (TNR) can be calculated by Equations (12) and (13):(12)$$TPR=\frac{TP}{TP+FN}\times 100$$
(13)$$TNR=\frac{TN}{TN+Fp}\times 100$$

The precision and recall values can be calculated by Equations (14) and (15) [55]:(14)$$PPV=\frac{TP}{TP+FP}\times 100$$
(15)$$Recall=\frac{TP}{TP+FN}\times 100$$

F-score can measure a harmonic mean of precision and recall, which can be computed by Equation (16):(16)$$F\_score=\frac{2TP}{2TP+FP+FN}\times 100$$

The false discovery rate (FDR) quantifies the proportion of unnecessary alerts, which can be calculated by Equation (17):(17)$$FDR=\frac{FP}{FP+TP}\times 100$$

The False Negative Rate (FNR) can be calculated by Equation (18):(18)$$FNR=\frac{FN}{FN+TP}\times 100$$

## 5. Results and Discussion

This section explores the gait recognition efficiency of the ABDGNet model and compares it with popular traditional CNN algorithms applied to various gait datasets, including LeNet, AlexNet, VggNet, Inception, Resnet50, Xception, and ABDGNet. The performance of each algorithm is analyzed in detail. These algorithms, with their predefined layers and parameters, utilize the settings shown in Table 3 for the ABDGNet model. Key evaluation metrics such as accuracy, sensitivity, specificity, recall, F-score, training time, and FNR are used to assess the quality of the traditional algorithms and to facilitate a comparison with the proposed algorithm.

To evaluate the proposed models, we used the following approaches. Firstly, all datasets used were split by (the train-test-split) model into three parts: train, test, and validation set, with a percentage of 70:15:15%, respectively.

Then, the proposed framework was compiled and fitted according to the number of the fine-tunned hyperparameters, which are listed in Table 4; a learning rate of 0.0005 to determine the step size for updates; the number of epochs, indicating model stability; a batch size set to 32, specifying the number of samples processed simultaneously; and a momentum factor of 0.5, used for calculating adaptive learning rates and maintaining a decaying average of past gradients. All tests were conducted on a PC with Microsoft Windows 10, a 7-core processor running at 4.0 GHz, 12 GB of RAM, and an NVidia Tesla 16 GB GPU.

### 5.1. LeNet

The performance of the LeNet algorithm was determined by training all tested gait datasets. The algorithm was trained using the following parameters: 32 × 32 image size, 30 epochs, and batch size 32. Table 5 details the results of the LeNet evaluation metrics. Figure 10 illustrates the precision curves for various classes. LeNet’s accuracy was estimated to be approximately 92.6%, with a total runtime of nearly 1 min and 39 s. Additionally, Figure 11 displays the LeNet model’s loss and accuracy curves.

### 5.2. AlexNet

The AlexNet model was tested by using gait datasets to evaluate its performance. Training and fitting the model was applied using the following parameters: 227 × 227 image size, 30 epochs, and batch size 32. Table 6 illustrates the evaluation metrics results of AlexNet. Figure 12 shows the AlexNet model loss and accuracy curves, which achieved a 99.1% accuracy value in a total training time of 4 min 26 s. Figure 13 shows precision–recall curves.

### 5.3. VggNet

Training all tested gait datasets determined the competence of VGG modules (VGG16 and VGG19). Using an image size of 224 × 224, 30 epochs, and a batch size of 32, two VGG models were built. Although both architectures produced similar performance outcomes, one took less time to run than the other; Table 7 shows the performance results of VggNet. The loss and accuracy curves of the VggNet model, which achieved 98.7% accuracy in approximately 41 min and 26 s, are shown in Figure 14. Figure 15 displays the VggNet model’s precision–recall curves.

### 5.4. Inception-v3

The Inception model was trained on gait datasets with specific parameters: 224 × 224 RGB image size, 30 epochs, and batch size 32. Table 8 shows the model’s evaluation metrics. Figure 16 displays the loss and accuracy curves of the Inception model, which achieved 96.8% accuracy in 37 min and 36 s. The precision–recall curves for various classes are depicted in Figure 17.

### 5.5. Resnet

The performance of the ResNet50 model was established by training all tested gait datasets. Fitting the model was developed using the following parameters: 224 × 224 RGB image size, 30 epochs, and batch size 32. Table 9 displays ResNet50 evaluation metrics results. ResNet accuracy was around 98.7%, with a total training time of nearly 46 min and 17 s. Figure 18 illustrates the classes’ precision–recall curves of ResNet. Figure 19 shows the loss and accuracy curves.

### 5.6. Xception

After fitting the Xception model, all tested gait datasets’ performance metrics were evaluated using 224 × 224 RGB image size, 30 epochs, and a batch size of 32. Table 10 displays the evaluation metrics results of the Xception model. Figure 20 illustrates the Xception model loss and accuracy curves. The Xception model achieved an accuracy of 96.4% in approximately one and a half hours. Class precision curves for the Xception model are depicted in Figure 21.

The performance results of the traditional CNN indicate that gait recognition values range between 92 and 99%, albeit with high running times. ABDGNet was also applied to the same dataset, aiming to enhance these performance metrics.

### 5.7. The Proposed CNN (ABDGNet)

The proposed CNN model (ABDGNet) was compared to several traditional CNN and existing models using three publicly available gait datasets: CASIA, OU-ISIR, and OU-MVLP. It achieved high accuracy rates, reaching 99.9% for CASIA-(B) and OU-ISIR and 99.7% for OU-MVLP in a relatively short training time, as shown in Table 11. Figure 22 displays the loss and accuracy curves of various applied datasets.

### 5.8. Evaluation of CASIA

The proposed gait recognition model was evaluated using the CASIA dataset according to the optimal parameters listed in Table 3. The comparison between the CASIA-B dataset and the traditional CNN was conducted, as detailed in Table 12 and Table 13. It was then extended with four existing models, including [11,56,57], as illustrated in Table 14. The average value of results [54] was calculated according to Equation (19):(19)$$MSE=\sum_{i=1}^{M}\sum_{j=1}^{N}|x_{ij}-y_{ij}{|}^{2}$$

The results indicated the effectiveness of the proposed model. Table 13 highlights that the proposed model outperformed the traditional CNN models in accuracy and training time, achieving 99.9% accuracy within three minutes. Figure 23 complements this by showing the training times for all traditional CNNs. Furthermore, Table 12’s data reveal the superior performance of the proposed model over other existing approaches, achieving a 100% recognition rate across all angles. Notably, for angles within the range of 36°, 108°, and 180°, the model maintained a high recognition rate of 99.0%.

### 5.9. Evaluation of OU-ISIR

The proposed ABDGNet model was also evaluated using the OU-ISIR dataset, following the same optimal parameters. According to Table 15, ABDGNet, when benchmarked against existing models such as those in references [19,22], showcased superior performance. It achieved a notable 99.9% accuracy in gait recognition within approximately three minutes. This result places ABDGNet ahead of current state-of-the-art models in the field.

### 5.10. Evaluation of OU-MVLP

The evaluation of ABDGNet using the OU-MVLP dataset, which comprises data from 14 different view angles, confirms the model’s effectiveness. Adhering to optimal parameters, ABDGNet’s performance was compared with methods from references [17,58,59], as detailed in Table 16. This comparison reveals that ABDGNet surpasses the existing models, achieving a remarkable recognition accuracy of 99.8% in a notably short time frame of approximately three minutes.

The research findings are divided into two phases. In the first phase, a comparative analysis was conducted between the proposed convolutional neural network (CNN) model and traditional CNNs. Table 13 summarizes the results, showing that ABDGNet achieved precision and accuracy of 99.6% and 99.9%, respectively, within 3 min and 25 s. Figure 23 illustrates the training times for traditional CNN models and the proposed ABDGNet. The proposed model achieved the highest precision of 99.6% and an accuracy of 99.9%, all within a brief training duration of only 3 min and 25 s.

Multi-scale image processing has demonstrated its pivotal role in enhancing gait recognition accuracy by accommodating variations in distance and perspective, which are natural in gait data. Furthermore, utilizing multi-scale approaches instead of a single one can improve the feature representation, resulting in better differentiation between various classes and enhancing object localization within an image. In our analysis, when the model applied a single-scale approach with image sizes of 50 × 50, 100 × 100, and 150 × 150 pixels, it achieved accuracy values of 98.5%, 98.8%, and 99.0%, respectively. In contrast, the multi-scale approach consistently showed superior performance, with accuracy rates reaching 99.9% for the CASIA dataset and 99.8% for the OU-MVLP dataset.

This enhancement is attributable to the multi-scale approach’s capacity to capture and integrate features across different resolutions, thereby improving the model’s robustness and adaptability to varying real-world conditions. The superior performance of the multi-scale approach underscores its effectiveness in handling the inherent variability in gait patterns more efficiently than the single-scale approach.

Furthermore, a detailed comparative study incorporating other recent gait recognition studies using CNN algorithms, such as ABDGNet, is presented in Table 17. The comparative results reinforce the superior accuracy and efficiency of the proposed multi-scale approach over other contemporary methods. ABDGNet’s performance metrics stand out in this comparison, highlighting its superiority over other CNN-based gait recognition methods.

## 6. Conclusions

This paper introduced ABDGNet, an advanced deep learning model that targets improvements in gait recognition. Utilizing a specialized CNN architecture, ABDGNet carefully processes, segments, and enhances gait images, effectively handling multiple scales of input data. An in-depth comparative analysis against traditional CNN models, based on essential performance metrics, has confirmed the enhanced capability of ABDGNet. Notably, the model demonstrates outstanding performance with an accuracy of 99.9%, a precision of 99.6%, a recall of 100%, a specificity of 99.0%, and a remarkably low False Negative Rate of 0.36. In addition to its precision, ABDGNet distinguishes itself with its efficient processing, evidenced by reduced training times compared to its counterparts. These findings underscore ABDGNet’s substantial advancements in speed and accuracy, indicating a significant leap forward in gait recognition technology. The study’s results advocate for ABDGNet as a robust and reliable solution, offering considerable promise for future surveillance and biometric authentication applications.

## Figures and Tables

**Figure 1 biomimetics-09-00364-f001:**
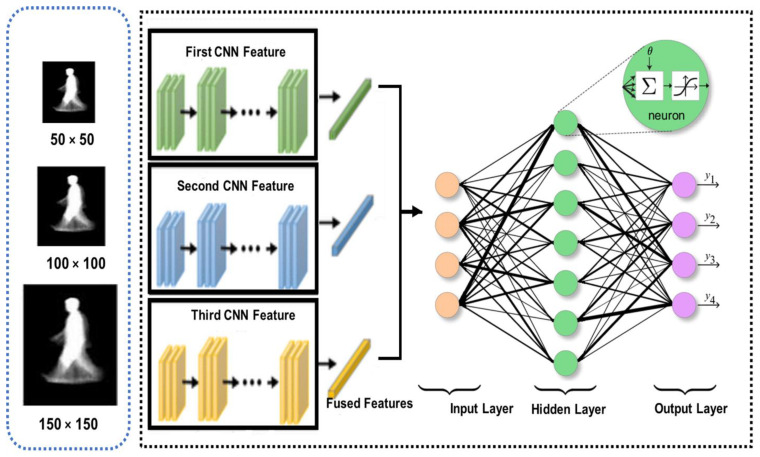
Gait recognition approach components.

**Figure 2 biomimetics-09-00364-f002:**
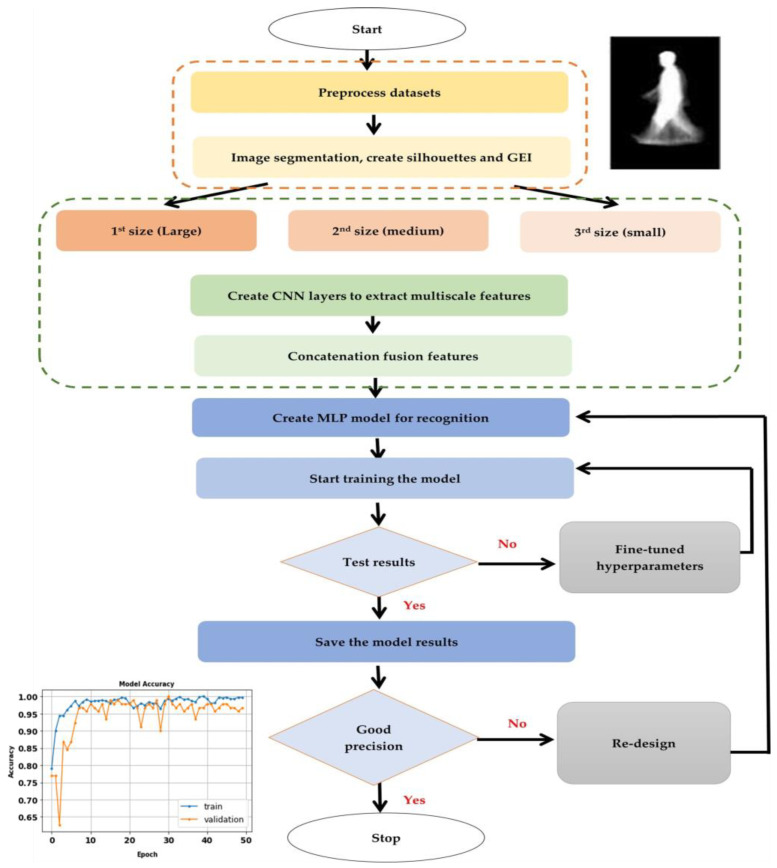
The proposed model flowchart.

**Figure 3 biomimetics-09-00364-f003:**
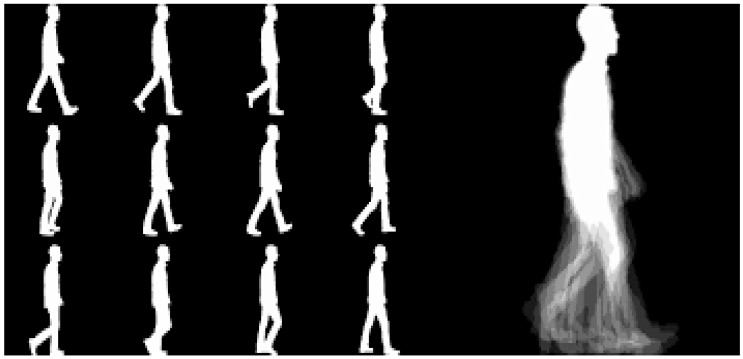
Gait energy image (GEI) for one person.

**Figure 4 biomimetics-09-00364-f004:**
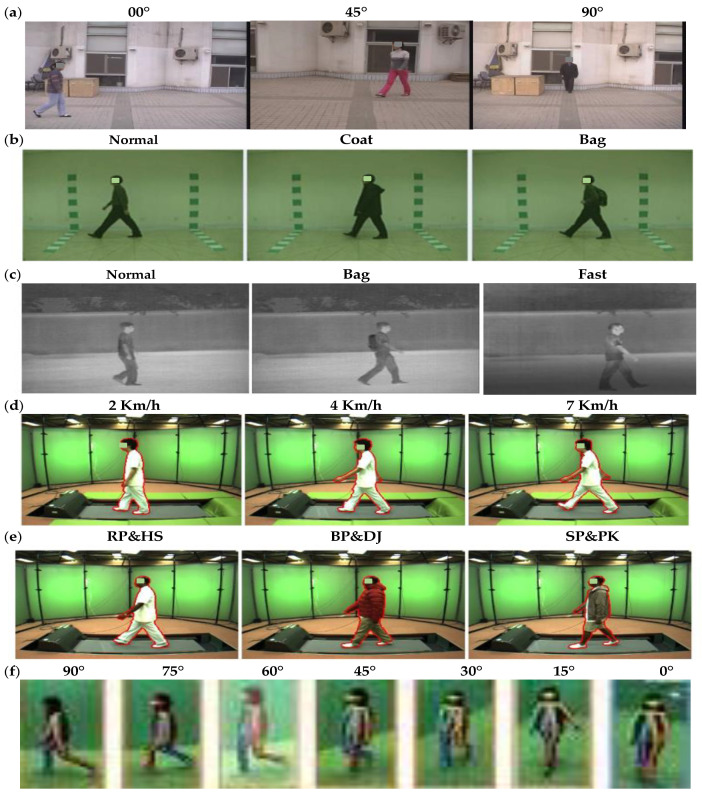
Sample of class label for (**a**) CASIA (A), (**b**) CASIA (B), (**c**) CASIA (C), (**d**) OU-ISIR (A), (**e**) OU-ISIR (B), and (**f**) OU-MVLP datasets.

**Figure 5 biomimetics-09-00364-f005:**
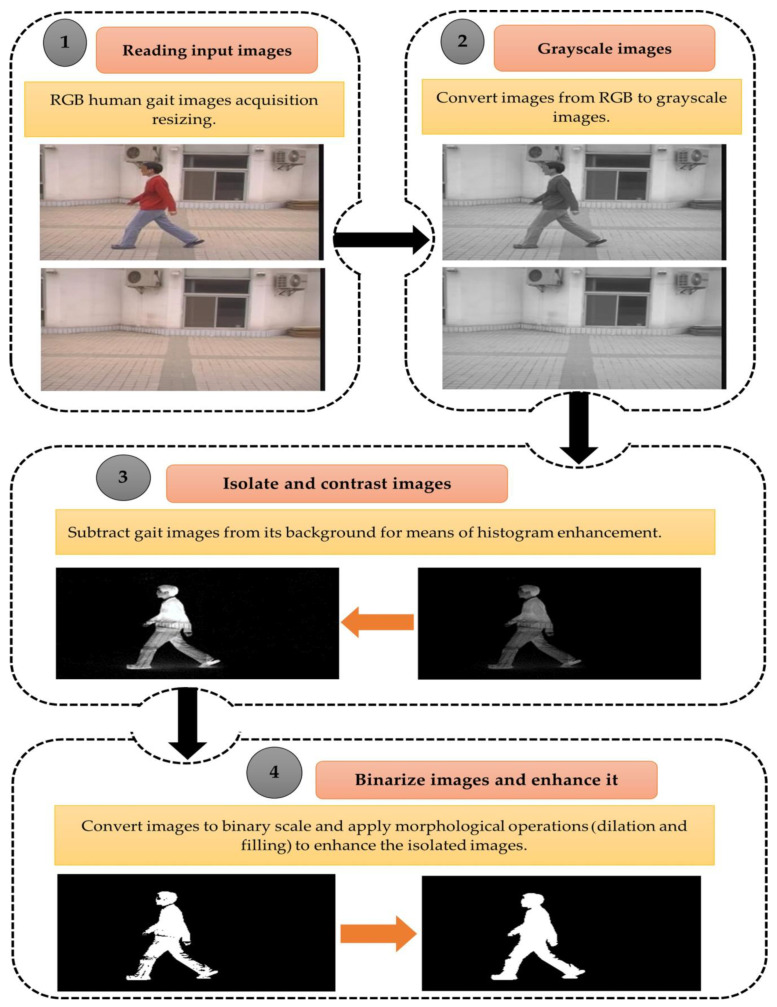
Gait dataset preprocessing steps.

**Figure 6 biomimetics-09-00364-f006:**
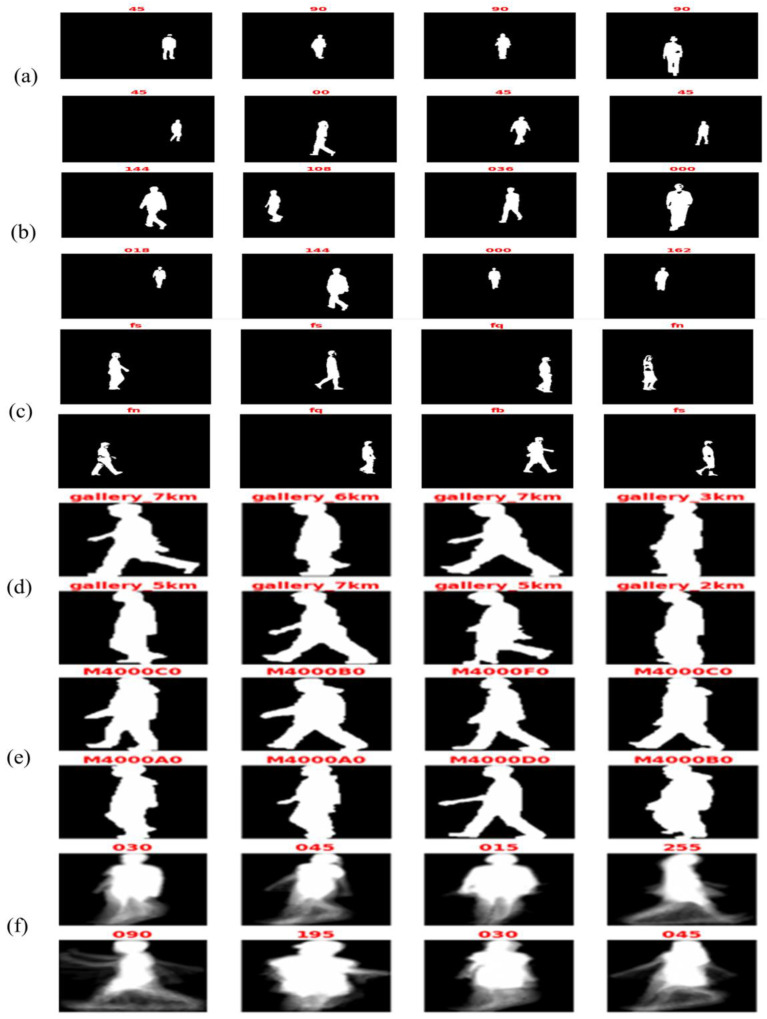
Sample of the preprocessed (**a**) CASIA (A), (**b**) CASIA (B), (**c**) CASIA (C), (**d**) OU-ISIR-A, (**e**) OU-ISIR-B, and (**f**) OU-MVLP gait datasets.

**Figure 7 biomimetics-09-00364-f007:**
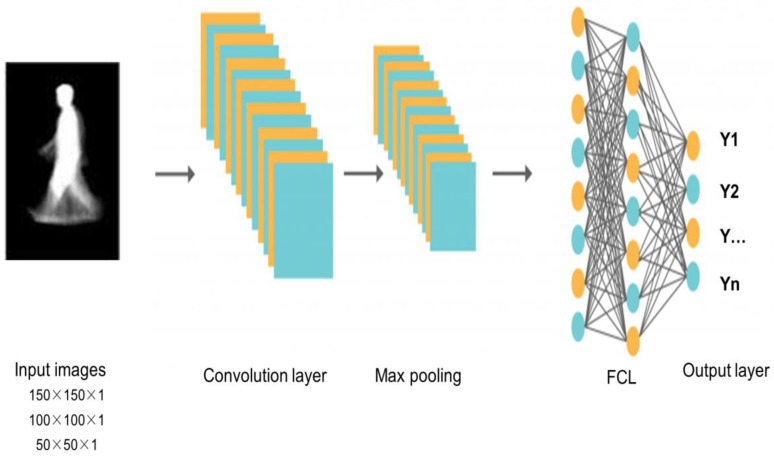
The ABDGNet architecture.

**Figure 8 biomimetics-09-00364-f008:**
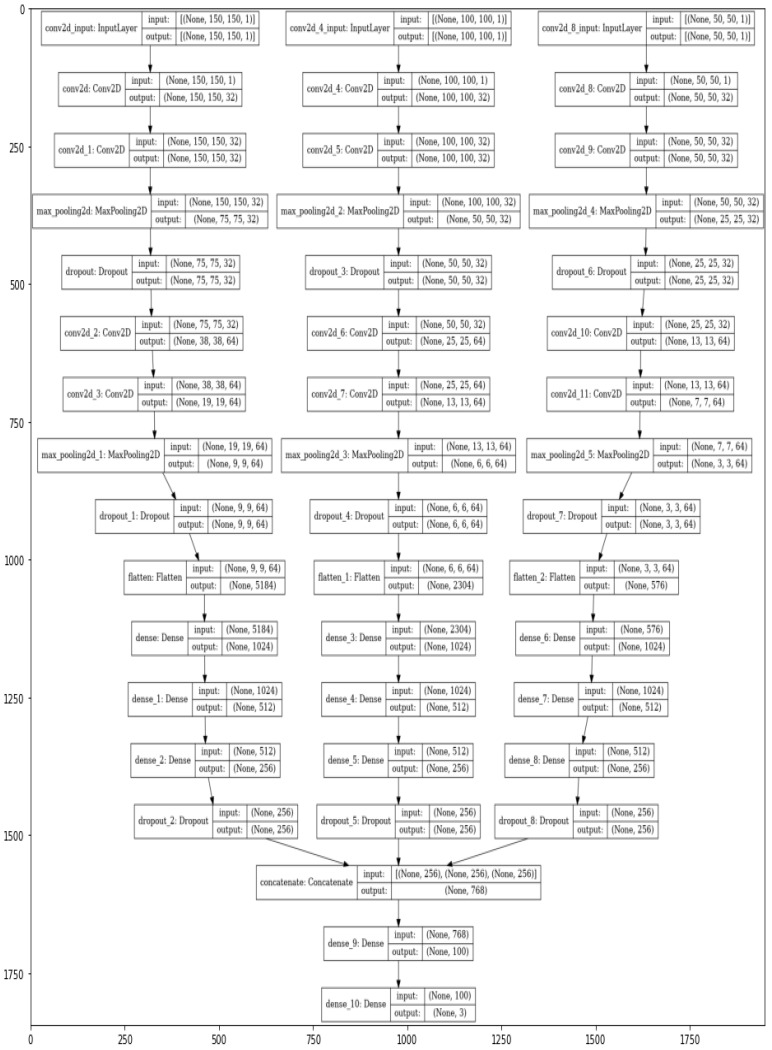
The ABDGNet layers using CASIA dataset.

**Figure 9 biomimetics-09-00364-f009:**
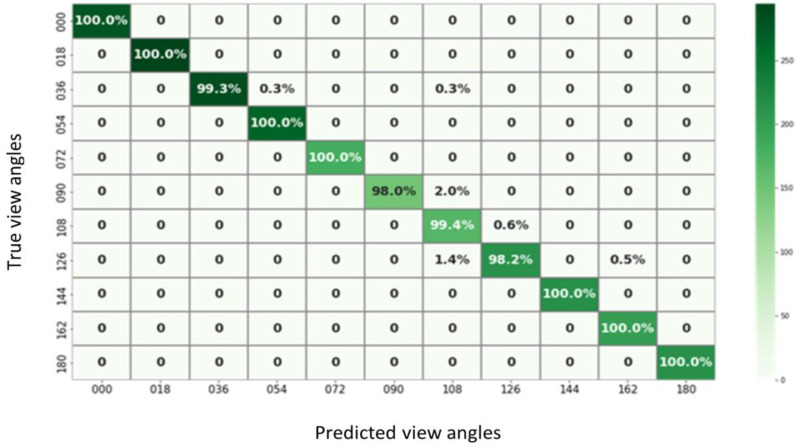
The ABDGNet confusion matrix.

**Figure 10 biomimetics-09-00364-f010:**
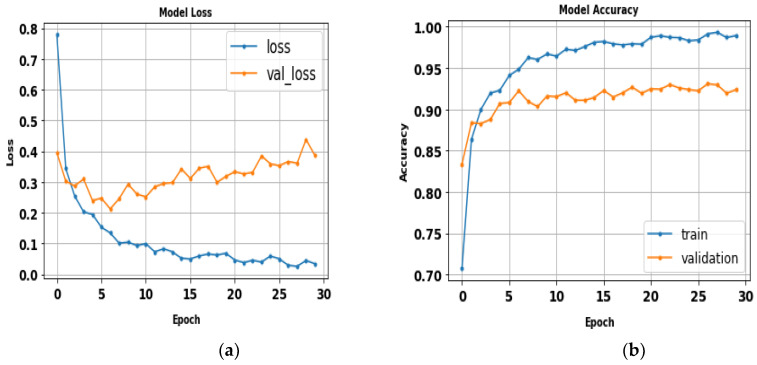
(**a**) Loss curve; (**b**) accuracy curve of LeNet.

**Figure 11 biomimetics-09-00364-f011:**
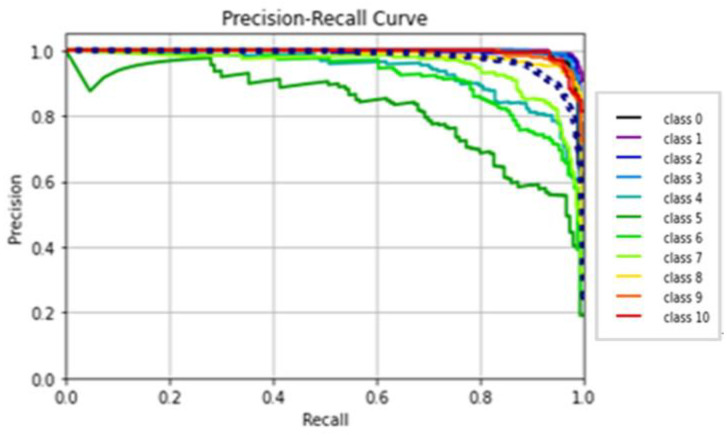
Classes’ precision–recall curves of LeNet.

**Figure 12 biomimetics-09-00364-f012:**
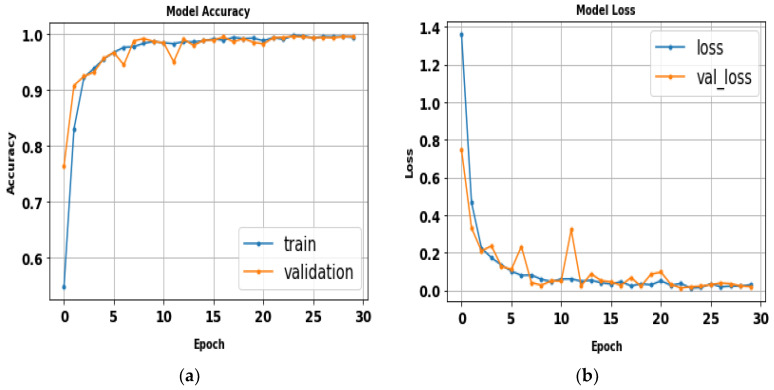
(**a**) Loss curve; (**b**) accuracy curve of AlexNet.

**Figure 13 biomimetics-09-00364-f013:**
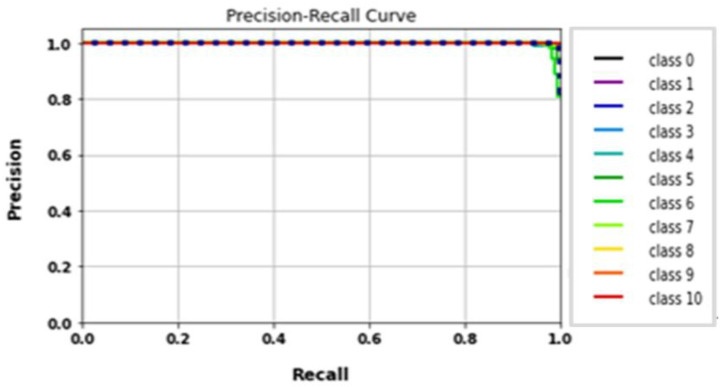
Classes’ precision–recall curves of AlexNet.

**Figure 14 biomimetics-09-00364-f014:**
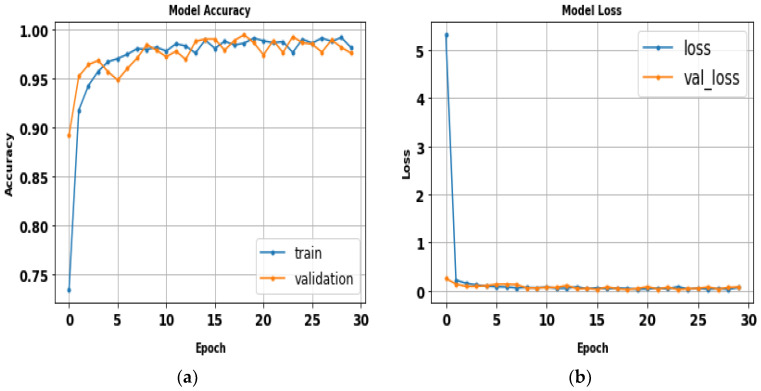
(**a**) Loss curve; (**b**) accuracy curve of VggNet.

**Figure 15 biomimetics-09-00364-f015:**
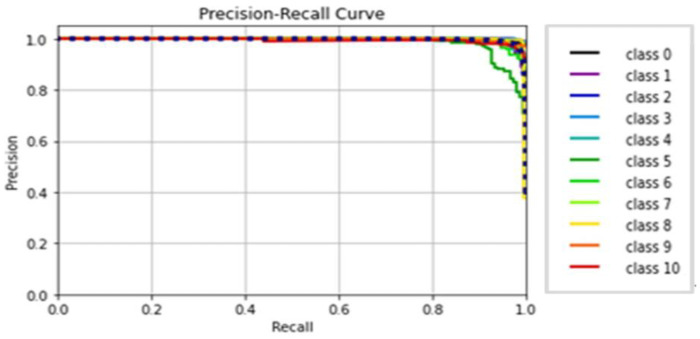
Classes’ precision–recall curves of VggNet.

**Figure 16 biomimetics-09-00364-f016:**
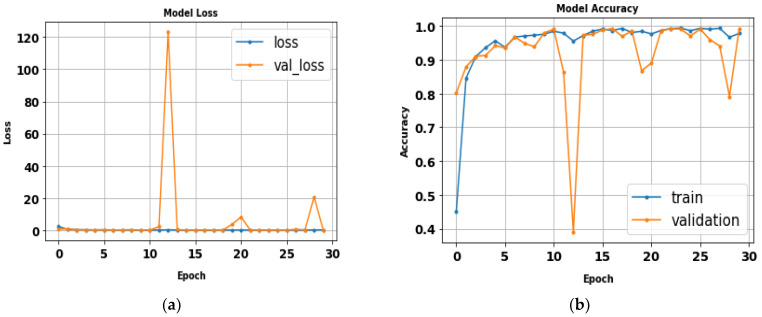
(**a**) Loss curve; (**b**) accuracy curve of Inception-v3 model.

**Figure 17 biomimetics-09-00364-f017:**
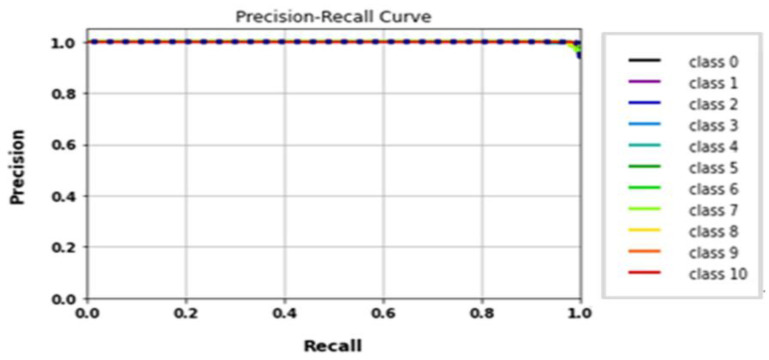
Classes’ precision–recall curves of Inception model.

**Figure 18 biomimetics-09-00364-f018:**
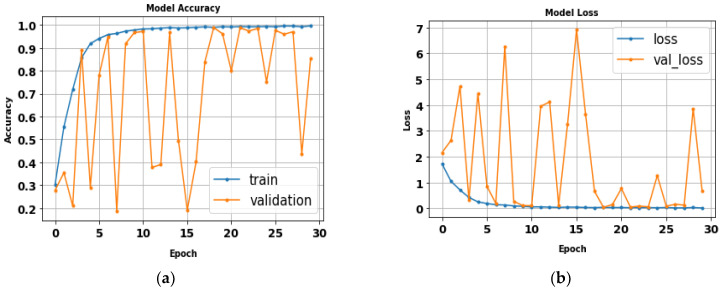
(**a**) Loss curve; (**b**) accuracy curve of ResNet model.

**Figure 19 biomimetics-09-00364-f019:**
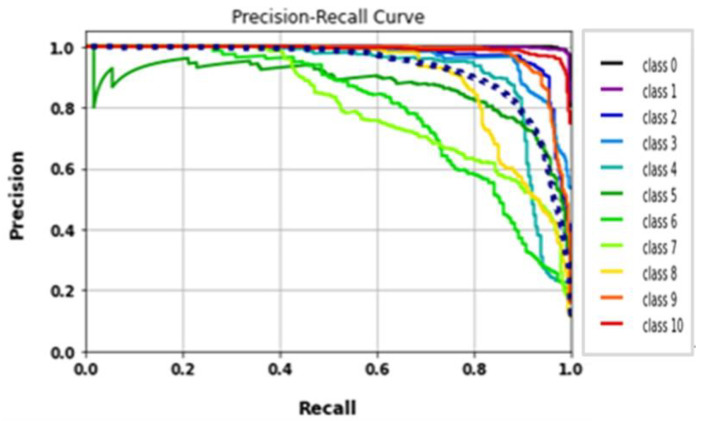
Classes’ precision–recall curves of ResNet model.

**Figure 20 biomimetics-09-00364-f020:**
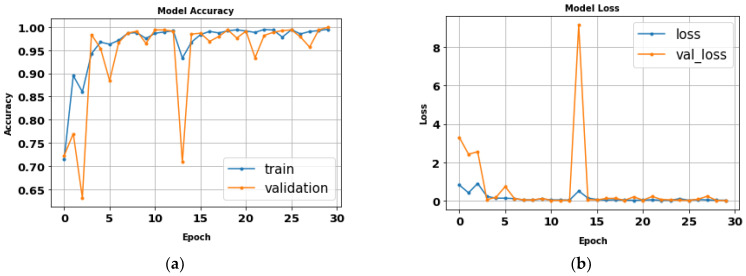
(**a**) Loss curve; (**b**) accuracy curve of Xception model.

**Figure 21 biomimetics-09-00364-f021:**
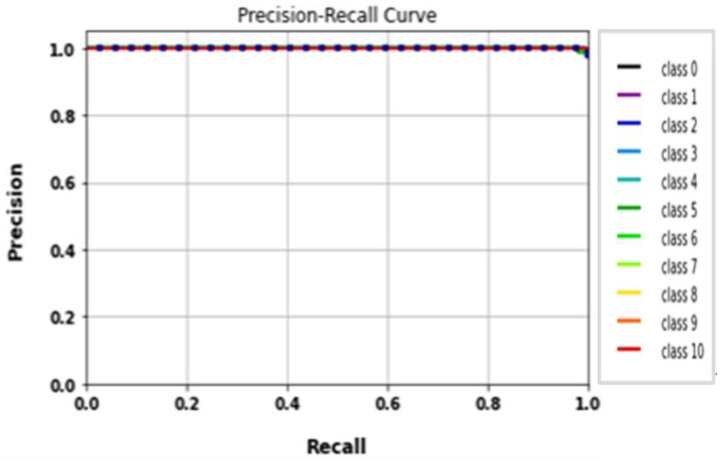
Classes’ precision–recall curves of Xception model.

**Figure 22 biomimetics-09-00364-f022:**
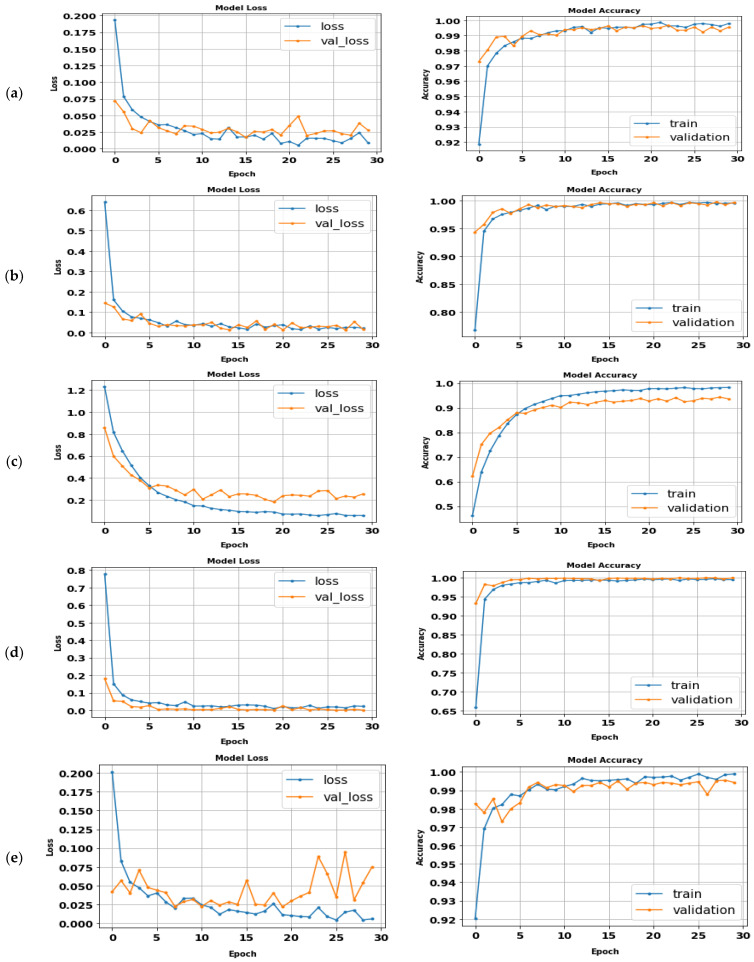
The ABDGNet loss and accuracy curves for (**a**) CASIA (A), (**b**) CASIA (B), (**c**) OU-ISIR-A, (**d**) OU-ISIR-B, and (**e**) OU-MVLP.

**Figure 23 biomimetics-09-00364-f023:**
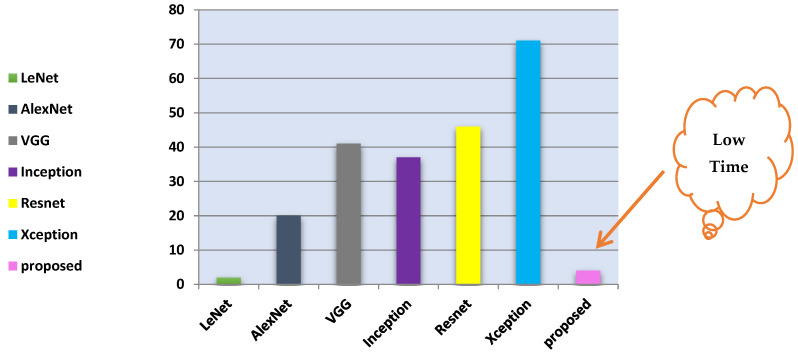
Comparison of training time for various CNN models.

**Table 1 biomimetics-09-00364-t001:** Comparison between the various state-of-the-art techniques for gait recognition.

Study	Dataset	Goal	Methodology	Challenges
Liao et al. [16]	CASIA [31] OU-ISIR [32]	This study introduced a model for resolving the problem of large-angle intervals.	They have introduced a new view synthesis method based on view space coverage.	Do not deal with real datasets.
Omar et al. [17]	CASIA [31] OU-ISIR [32] OU-MVLP [33]	This research proposed a gait authentication model.	They generated GEIs from gait images. Then, they estimated the angle of the gait and recognized it using a convolutional neural network.	Do not enhance the segmented datasets.
Xinhui et al. [19]	CASIA [31] OU-ISIR [32]	This paper presented a human gait identification algorithm from a video sequence frames.	They extracted the discriminative features in the temporal and spatial domains and recognized them.	Accuracy is very low.
K. Zhang et al. [22]	CASIA [31] OU-ISIR [32] OU-LP-Bag β [34]	This study presented a novel gait authentication model.	They introduced a novel Joint Unique-gait and Cross-gait Network (JUCNet) representation to incorporate the benefits of both approaches.	Complicated system.
Shiqi et al. [25]	CASIA [31]	This study presented a transform model.	They introduced the GaitGANv2 transform model, which transforms gait images from any viewpoint to a side view and then recognizes it.	Accuracy is very low.

**Table 2 biomimetics-09-00364-t002:** Gait data description.

Data	Image Size	Category	Category/Image
CASIA-A	240 × 320	3	3500–6000
CASIA-B		11	1200–2500
CASIA-C		4	3000–5000
OU-ISIR (A)	128 × 88	9	2000–3000
OU-ISIR (B)		8	2500–2800
OU-MVLP		14	1300–1500

**Table 3 biomimetics-09-00364-t003:** Parameter settings for ABDGNet model.

Parameters	Value
Image Size	150 × 150
	100 × 100
	50 × 50
Mini Batch Size	32
Initial Learning Rate	0.0005
Number of Epochs	30
Optimizer	Adam
Momentum Factor	0.5
Execution Environment	GPU
CNN Initial Weights	ImageNet
Activation Function	SoftMax (final dense layer)

**Table 4 biomimetics-09-00364-t004:** Parameter settings for traditional pre-trained models.

Parameters	Value
Image Size	LeNet = 32 × 32
	AlexNet = 227 × 227
	VGGnet = 224 × 224
	Inception-v3 = 224 × 224
	ResNet = 224 × 224
	Xception = 224 × 224
Mini Batch Size	32
Initial Learning Rate	0.0005
Number of Epochs	30
Momentum Factor	0.5
Execution Environment	GPU
CNN Initial Weights	ImageNet
Activation Function	SoftMax (final dense layer)

**Table 5 biomimetics-09-00364-t005:** LeNet evaluation results on key performance metrics.

Model_1	Sens (%)	Recall	F-Score	Spec (%)	FNR (%)	Training Time
LeNet	92.6	0.91	0.92	99.2	7.73	1 min 39 s.

**Table 6 biomimetics-09-00364-t006:** AlexNet evaluation results on key performance metrics.

Model_2	Sensitivity	Recall	F-Score	Specificity	FNR	Training Time
AlexNet	0.994	0.99	0.99	0.999	0.526	40 min 26 s

**Table 7 biomimetics-09-00364-t007:** The VggNet evaluation metrics results.

Model_3	Sensitivity	Recall	F-Score	Specificity	FNR (%)	Time
VggNet	0.987	0.99	0.98	0.998	1.21	41 min 26 s

**Table 8 biomimetics-09-00364-t008:** Performance evaluation metrics for the Inception model.

Model_4	Sensitivity	Recall	F-Score	Specificity	FNR (%)	Training Time
Inception-v3	0.968	0.96	0.96	0.996	3.1	37 min 36 s

**Table 9 biomimetics-09-00364-t009:** Performance evaluation metrics for the ResNet50 model.

Model_5	Sensitivity	Recall	F-Score	Specificity	FNR (%)	Training Time
ResNet50	0.987	0.99	0.98	0.998	1.24	46 min 17 s

**Table 10 biomimetics-09-00364-t010:** Performance evaluation metrics for the Xception model.

Model_6	Sensitivity	Recall	F-Score	Specificity	FNR (%)	Training Time
Xception	0.964	0.96	0.96	0.996	3.56	1 h 11 min 23 s

**Table 11 biomimetics-09-00364-t011:** The proposed (ABDGNet) performance results.

Dataset Name	Acc (%)	Precision (%)	Recall (%)	F-Score	Specificity (%)	FNR (%)	Training Time
CASIA (A)	99.6	99.4	99	0.99	99.7	0.55	3 min 51 s
CASIA (B)	99.9	99.6	100	0.99	99.9	0.36	3 min 25 s
CASIA (C)	95.3	90.7	90	0.907	96.9	9.25	4 min 14 s
OU-ISIR A	97.4	88.7	87	0.89	98.5	11.2	3 min 30 s
OU-ISIR B	99.9	99.8	100	0.99	99.9	0.16	3 min 27 s
OU-MVLP	99.8	98.6	98	0.98	99.8	1.31	3 min 13 s

**Table 12 biomimetics-09-00364-t012:** Classes’ precision of all implemented algorithms.

Class	LeNet	AlexNet	VggNet	Inception	ResNet	Xception	Proposed Model
000	97.51	100	100	100	99.75	100	98.93
018	97.99	100	100	100	100	99.66	100
036	96.90	100	100	99.60	100	99.65	99.65
054	95.27	99.62	100	100	99.54	99.62	100
072	86.19	98.89	98.90	97.80	99.26	99.45	100
090	74.83	94.87	96.08	98.60	93.52	98.67	98.03
108	81.87	99.39	93.71	100	93.56	99.40	99.40
126	85.58	100	98.55	99.50	99.69	100	100
144	94.55	96.85	97.69	98.10	99.39	100	100
162	98.38	100	99.49	100	98.02	100	100
180	96.35	99.53	99.07	100	100	100	99.53
Average	91.40	99.01	98.40	99.40	98.40	99.00	99.60

**Table 13 biomimetics-09-00364-t013:** Performance results.

Model	Accuracy	Precision	Recall	F1-Score	Specificity	FNR	Training Time
LeNet	0.91	0.93	0.91	0.92	0.99	7.73	1 min 39 s
AlexNet	0.99	0.99	0.99	0.99	1.00	0.53	20 min 26 s
VggNet	0.98	0.99	0.99	0.98	1.00	1.21	41 min 26 s
Inception	0.99	0.97	0.96	0.96	0.99	3.10	37 min 36 s
ResNet	0.98	0.99	0.99	0.98	0.99	1.24	46 min 17 s
Xception	0.99	0.96	0.96	0.96	0.99	3.56	1 h 11 min 23 s
Proposed	0.99	0.99	1.00	0.99	1.00	0.40	3 min 25 s

**Table 14 biomimetics-09-00364-t014:** Detailed analysis of view angles used in various studies on the CASIA dataset.

Model	0°	18°	36°	54°	72°	90°	108°	126°	144°	162°	180°	Average
[56]	0.93	0.92	0.90	0.92	0.87	0.95	0.94	0.95	0.92	0.90	0.90	0.92
[11]	0.95	0.96	0.95	0.96	0.95	0.97	0.97	0.94	0.96	0.97	0.97	0.96
[57]	0.43	0.78	0.99	-	0.98	0.82	0.77	0.76	0.57	0.42	0.35	0.69
[17]	0.94	0.95	0.97	0.97	0.98	0.98	0.98	0.98	0.97	0.95	0.93	0.96
Proposed	0.98	1.00	0.99	1.00	1.00	0.98	0.99	1.00	1.00	1.00	0.99	0.99

**Table 15 biomimetics-09-00364-t015:** Accuracy comparison of different gait recognition methods on the OU-ISIR dataset.

Reference	Accuracy
[19]	67.0%
[22]	77.6%
[17]	98.7%
[29]	95.0%
[41]	91.5%
Proposed	99.9%

**Table 16 biomimetics-09-00364-t016:** Performance comparison of various models across different view angles on the OU-MVLP dataset.

Model	0°	15°	30°	45°	60°	75°	90°	180°	195°	210°	225°	240°	255°	270°
[58]	0.79	0.89	0.93	0.95	0.95	0.95	0.95	0.96	0.90	0.95	0.95	0.93	0.94	0.94
[59]	0.79	0.87	0.89	0.90	0.88	0.88	0.87	0.81	0.86	0.89	0.89	0.87	0.87	0.86
[17]	0.93	0.95	0.95	0.97	0.98	0.97	0.98	0.92	0.94	0.95	0.95	0.97	0.97	0.98
Proposed	0.96	0.99	0.99	0.99	0.98	0.98	0.99	0.95	0.99	0.99	0.98	0.98	0.98	0.98

**Table 17 biomimetics-09-00364-t017:** Comparative analysis of the ABDGNET model with contemporary gait recognition studies.

Study	Dataset Used	Success Accuracy
Liao et al. [16]	CASIA [31] OU-ISIR [32]	The output features enhanced the gait recognition
Omar et al. [17]	CASIA [31] OU-ISIR [32] OU-MVLP [33]	99.1% 98.7% 98.4%
Xinhui et al. [19]	CASIA [31] OU-ISIR [32]	87.2% 67.0%
K. Zhang et al. [22]	CASIA [31] OU-ISIR [32] OU-LP-Bag [34]	93.2% 77.6% 79.3%
Shiqi et al. [25]	CASIA [31]	62.8%
Proposed	CASIA (A) CASIA (B) OU-ISIR (A) OU-ISIR (B) OU-MVLP	99.9% 99.6% 97.4% 99.9% 99.8%

## Data Availability

Data are available upon request. The preprocessing steps of our study Python code in GitHub at the following link: https://github.com/remooooooooooo/preprocessing-steps/blob/main/python (accessed on 20 April 2024).

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
