# Peer review of "A Novel Multi-Scaled Deep Convolutional Structure for Punctilious Human Gait Authentication"

_biomimetics, 2024, doi:10.3390/biomimetics9060364_

Round 1
Reviewer 1 Report
Comments and Suggestions for Authors
The topic is very interesting a current. The gait recognition is particularly used in order to realize the continuous authentication. The manuscript is satisfactory complete. The introduction describes the problem and introduces the proposed method. The related works section is complete. The proposed method and the experimental results are describe and commented in exhaustive way. The metrics used are correct and the obtained results on public databases are satisfactory. This reviewer have not suggestions for authors.
Comments on the Quality of English LanguageThe quality of the English language used is satisfactory, I suggest a re-reading to further refine the fluency.
Reviewer 2 Report
Comments and Suggestions for Authors
The paper is interesting but needs intensive revisions and formatting.
The introduction should clearly emphasize the problem addressed and the actual contributions in this paper.
It is too early in the manuscript to show the design of the proposed solution (Figure 1).
The related work section should highlight the research gap and include a technical comparison of the listed methods. Comparison criteria in the related work section should be used to compare the listed research works.
The Algorithms are presented in a non-professional format. Please use standard algorithmic forms to represent the algorithms mentioned in the paper. The conceptual design of the proposed method is missing. It is unclear which dataset has been used in this study! There is a confusing description of several datasets.
The presentation of the experimental work and the results must be improved to make the paper more readable. What is the goal of Table 17?
All figures are blurred, which must be presented in a high-resolution format.
References must be checked. Some references have double numbers, for example, 71. [71] and 72. [72] ??
Comments on the Quality of English LanguageThe paper should be checked by a native English speaker.
Reviewer 3 Report
Comments and Suggestions for Authors
The authors propose a human gait recognition method that uses silhouette images at multiple scales as input to a Convolutional Neural Network (CNN). The concept of using images of the same object at multiple scales as input to a CNN is not new and has been explored in other areas such as object recognition and scene recognition.
The novelty of the proposed approach is that, to my knowledge, the authors are the first to have explored this approach for gait recognition.
My comments to the authors and suggestions to improve the work are as follows:
The article should be significantly shortened. Among other things, it is not necessary to present all the details of the traditional algorithms, e.g. in section 3 on Convolutional Neural Network Models. Table 17 is not necessary. The comparison between different biometric methods and what makes gait recognition interesting should be limited to the introduction.
The text is difficult to follow. For example, the description of the algorithms presented is not clearly separated. There are overlaps in the description of the algorithms. For example, Algorithm 2 describes pre-processing steps that are also described in Algorithm 1. There are clear redundancies, a lack of modularity and an inconsistent presentation. For example, in the description of Algorithm 2, the terminology used to describe the output at the beginning does not match the returned values from the last step of the algorithm.
The study lacks the ablation study to evaluate the individual effects of the different components (such as layers, activation functions, etc.) on the performance of the model or to provide a rationale for the choice of components and parameters of the network.
To evaluate the contribution of the multi-scale approach to recognition accuracy and the performance of the proposed approach in general, the authors should compare the 3-scale approach specifically with a single-scale approach using the same preprocessing and model.
In section 4.6 of the performance evaluation, the confusion matrix is presented without any information about the inputs, i.e. the testing set. What do the labels shown in the confusion matrix mean and how do they relate to the Casia B dataset?
There is no information on whether the data in the training, test and validation sets are balanced.
Many other small improvements and/or corrections should be made throughout the paper and the presentation can be improved in many places. Some examples (there are many more):
The first paragraph in section 4 is unclear (the text in lines 247-248 should be improved)
- Why is the tracking of COVID-19 relevant?
- Past and present tenses are sometimes mixed in the same paragraph.
- The sentences in lines 154 and 155 are unclear and make no sense
- Figure 1 - the resolution of the image is missing (100x100)
- bioinformatics -> biometrics
- etc.
The Related work section states that the models mentioned are listed and compared in Table 15. However, only 3 models are listed in Table 15.
Inconsistent and/or too many significant digits are used throughout the document when reporting accuracy (e.g. lines 108, 122, 168, etc.),
Why do you use the same number of epochs (30) in all cases? In Fig. 18, for example, this number does not seem appropriate.
Other suggestions for the authors:
The contributions of the paper mentioned in the introduction section are quite overlapping, e.g. "developing a deep CNN algorithm ..." / "Implementing a fine-tuned multilayer perceptron (MLP) for feature recognition." / "Developing an optimized convolutional neural network (CNN) model". The technique for extracting and resizing gait images at three scales is not new.
Please check formatting (uniform font size (affiliations), alignment and layout of figures, page breaks, missing bullets in bulleted lists (contributions in the introduction), missing characters ('%' in line 108), first letter of last name is not capitalized [line 153 etc.), efficiency vs. accuracy (line 141), etc.
Make the code for preprocessing images available in a public repository. This will improve the reproducibility of results and make your work visible.
Explain the artifacts and the low quality of the images after preprocessing (Fig. 6, row (a) Casia A).
The images in Fig. 6 appear vertically compressed (or horizontally stretched).
In Table 16. it is not necessary to highlight the results in red color.
The terms concatenation and fusing are used inconsistently and sometimes together.
In the related work section, the original sources of the concepts where the work was first proposed are sometimes not cited or acknowledged, e.g. for GEI the original work has not been cited, instead a self-citation is used in line 66 (which may be appropriate in some cases in addition to the reference to the original work).
Comments on the Quality of English LanguageThe English language is mostly appropriate.
Round 2
Reviewer 2 Report
Comments and Suggestions for Authors
Thank you for addressing my comments. The paper looks better now, but some minor issues should be fixed.
The paper should be checked by a native English speaker.
The algorithms should be in a professional format. Please review similar articles in biometrics
Some statements need further investigation. For example, the authors mentioned that gait recognition is unique before it is contactless, which I think is not accurate.
Comments on the Quality of English LanguageThe paper should be checked by a native English speaker.
Reviewer 3 Report
Comments and Suggestions for Authors
In this revised version, the authors have addressed most of my comments, and the paper has improved. However, I still have a few concerns outlined below. I also believe the authors should focus more on emphasizing the contribution of the multi-scale approach.
- In response to my request, the authors compared the multi-scale approach (an important contribution of the paper) to a single-scale approach by presenting results but provided no further commentary or discussion. The paper states that the proposed algorithm for the OU-MLDP dataset achieves 99.6% accuracy, while the single-scale approach for a size of 150 achieves 99% accuracy. However, the authors write "99%" without specifying decimal precision, making it unclear if this refers to 99.0% or a higher value. I previously commented on the inconsistent use of decimal places. Even assuming it was an innocent oversight (e.g., 99.0%), the authors should elaborate on the differences between the two approaches.
- Inconsistencies in the number of decimal places persist throughout the paper. In Table 12, sometimes zero, one, or two decimal places are used, which could be due to formatting or notation errors rather than rounding inconsistencies (e.g., 99.4 might represent 99.42 rather than 99.40). Table 13, uses one, two, or three decimal places, etc.
- The authors state, “We have removed our self-citation and alternated it with other latest references.” However, it's essential to attribute contributions to the original papers where the method was first proposed, not just the most recent works. Additionally, self-supervised gait recognition is not mentioned in the related works.
- Following my remark, In Figure 6, Row a, the authors replaced images containing artifacts with different samples that lack artifacts, instead of explaining the presence of the artifacts even after the preprocessing of the samples. If artifacts are present, the preprocessing should be improved or clarified.
- The authors still did not explain why they use the same number of epochs (30) in all cases? In Fig. 18, for example, this number does not seem appropriate.
